# Financial Catastrophism Inherent with Out-of-Pocket Payments in Long Term Care for Households: A Latent Impoverishment

**DOI:** 10.3390/ijerph17010295

**Published:** 2020-01-01

**Authors:** Raúl Del Pozo-Rubio, Isabel Pardo-García, Francisco Escribano-Sotos

**Affiliations:** 1School of Social Sciences, Sociosanitary Research Center, Castilla-La Mancha University (UCLM), 16071 Cuenca, Spain; raul.delpozo@uclm.es; 2School of Economics and Business Administration, Sociosanitary Research Center, Castilla-La Mancha University (UCLM), 02071 Albacete, Spain; francisco.esotos@uclm.es

**Keywords:** out-of-pocket payments, catastrophic long-term care payments, impoverishing, poverty impact

## Abstract

Background: Out-of-pocket (OOP) payments are configured as an important source of financing long-term care (LTC). However, very few studies have analyzed the risk of impoverishment and catastrophic effects of OOP in LTC. To estimate the contribution of users to the financing of LTC and to analyze the economic consequences for households in terms of impoverishment and catastrophism after financial crisis in Spain. Methods: The database that was used is the 2008 Spanish Disability and Dependency Survey, projected to 2012. We analyze the OOP payments effect associated to the impoverishment of households comparing volume and financial situation before and after OOP payment. At the same time, the extent to which OOP payment had led to catastrophism was analyzed using different thresholds. Results: The results show that contribution of dependent people to the financing of the services they receive exceeds by 50% the costs of these services. This expenditure entails an increase in the number of households that live below the poverty. In terms of catastrophism, more than 80% of households dedicate more than 10% of their income to dependency OOP payments. In annual terms, the catastrophe gap generated by devoting more than 10% of the household income to dependent care OOP payment reached €3955, 1 million (0.38% of GDP). Conclusion: This article informs about consequences of OOP in LCT and supplements previous research that focus on health. Our results should serve to develop strategic for protection against the financial risk resulting from facing the costs of a situation of dependence.

## 1. Introduction

Total spending on health care on average accounts for 9% of GDP in the OECD while long-term care (LTC) absorbs 1.5% of GDP [1,2]. Given the rates of world population ageing and dependence, this spending will continue to rise in the coming years [2] and put upward pressure on public finances, although some studies reveal the opposite [3]. Following the crisis of 2007, economic policies implemented a reduction in public spending in general, and in health and long-term care expenditure, in particular. Legislation was also modified as regards the distribution of the financial burden between the public sector and users, in the form, for example, of out-of-pocket (OOP) payments and insurance in the LTC field [4].

OOP payments associated with LTC are defined as beneficiary participation in the cost of a service. It has two functions: to gain efficiency from the use of an asset and/or service, curtailing excessive demand, known as moral hazard, which arises when the price paid by a consumer is lower than the marginal value or utility [5,6,7], and reducing costs by raising additional financial resources.

According to Xu et al. (2007) [8] many countries rely heavily on patients’ OOP payments to providers to finance their health care systems but there is widespread debate in public health systems on whether to establish a OOP payment or not, and the effects that OOP payment has on the use of services, trying to determine if there is a negative price elasticity between price and service use. There is extensive literature examining the impact of out-of-pocket expenditures for health services in overall terms [9,10,11,12] and also in specific segments of health services, including drugs [13], interventions [14], or specific diseases, such as cardiovascular diseases [15,16], HIV [17], cancer [18,19,20,21], or non-communicable diseases [22]. In their systematic review, Kiil and Houlrberg (2014) [23] found that OOP payment has a negative effect on demand, except in the case of hospitalizations, and reduces the use of prescription medicine. Others studies suggest that OOP payment increases were also found to lead to decreased utilization of services, including hospitalizations, physician visits, prescription drugs, and outpatient clinic visits [24]. Recently, Yerramilli, Fernández, and Thomson (2018) [25] revealed that a number of financial protection studies in Europe focus on middle-income countries, with obsolete information (the last year available in many countries was 2011). In addition, very few studies conduct an in-depth analysis of the effects of catastrophism (incidence and intensity) associated with OOP expenditure, and the factors associated with catastrophism and the likelihood of incurring such catastrophic payments [25].

OOP payments also impact on equity [26,27] join to catastrophe and impoverishment although the literature has devoted less attention to these issues. According to Antonanzas and Brías (2013) [28], few studies have been conducted on the effects of OOP on equity and no conclusions have been reached, although, for example, reduced use of prescription drugs in response to higher OOP payments by low-income adults on public assistance have also been reported in Canada [29] and in Spain [30]. Kiil and Houlberg (2014) [23] indicate that some studies find an association between low income and higher elasticity-price of demand, while others do not. In the same vein, González López-Valcárcel et al. (2016) [6] only locate three studies analyzing the effect of OOP payment on the use of health services by vulnerable population groups. 

Although the literature on catastrophe and impoverishment is also limited [31,32,33,34,35,36], the results show that the implementation of health care OOP payment puts many households at financial risk. Even a small payment can generate a financial problem in a poor household, forcing the reduction of other basic expenses such as food, hygiene, or the education of their children, which affects quality of life while families may fall into poverty or become poorer.

A review of the literature on OOP payment shows that its effects on health, although varying, are well documented. In the case of private health insurance systems, a number of studies in the United States analyze the effect of the lack of health insurance on the chronically ill [37,38,39,40,41] or people with disabilities [42,43,44,45,46]. However, those analyzing the effects of OPP payments or health insurance on LTC are more limited in number although interest in this issue is growing. Some studies have analyzed OOP payments for specific components of health care such as the chronically ill [47,48] and the disabled population [49,50,51]. In addition, a study across the OECD has measured the protection of long-term care and differences between countries. For example, in the United States, people with assets are expected to use them to pay for care until they become legally impoverished and eligible for social protection [52]. However, to the best of our knowledge, there is no study analyzing the catastrophic financial effect of OOP payment associated with LTC in any country.

In Spain, the LTC system is funded by revenue from taxes and user’s OOP payments. Specifically, the 2006 Act for the Promotion of Personal Autonomy and Care of Dependent Persons [53], commonly known as the Dependency Act (DA), sets out that users will contribute a third of the cost of the service, depending on their economic situation. The limited number of studies conducted in this regard [54,55] show that their contribution is far from this quantity, being between 28.5% and 21.3%. In 2012, against a backdrop of economic crisis and faced by the need to reduce public debt, the government amended the legislation on OOP payment, leading to user contributions to rise to 50% [56,57]. Table 1 shows the differences in the legislation before and after 2012 for OOP payments associated with LTC [58,59,60]. This legislative reform has had no effect on the demand for services because despite not being wholly insensitive to OPP payments, it is generally accepted that demand for LTC services is inelastic, and this can have an impact on equity. Although in health terms, as highlighted in various literature reviews, there is no clear causality between health and recession, economic crises increase poverty and have a comparatively greater impact on vulnerable populations such as the elderly [61,62,63].

Furthermore, the amendment implemented in 2012, in a situation of economic crisis, makes it more relevant to analyze the effect it has had in terms of impoverishment and catastrophe. OOP expenditure for LTC involves a significant financial burden for population whose ability to carry out the basic activities of daily living is restricted [51,64], increasing the risk of financial catastrophism for this vulnerable population group [65,66].

On the other hand, the original implementation of the DA was designed to be progressive, where people with level III dependency should have benefitted from the DA in 2007; those with level II should have benefitted in 2008–2011, and people with level I in 2012–2015. However, the structural legislation previously cited postponed the inclusion of people with moderate dependence (level I) until July 2015 [59,60].

The aim of this paper is to estimate the amount of OOP payment for LTC made by users in accordance with the 2012 legislation, and to analyze the effect of this amendment in terms of the impoverishment and catastrophism of households in Spain. The paper is organized as follows: In the second section, we present the databases used in the analysis and the methodology employed to calculate OOP payment, impoverishment, and catastrophic expenditure in households. The third section presents the empirical results. The final section is devoted to a discussion of the results obtained and the main conclusions.

## 2. Material and Methods

### 2.1. Sample

The Spanish Disability and Dependency Survey for 2008 (SDDS) conducted by the Spanish National Statistics Institute [67] was used to obtain the socioeconomic, demographic, and health profile and the characteristics of the environment of people with disabilities in Spain. Specifically, we used the households section of the SDDS, which contains surveys on 22,975 persons. The methodology of the survey assigns weights to each item so as to extrapolate the findings to the population with disabilities in Spain. Apart from information related to disabilities, impairments, and limitations, it also contains information on the income and financial situation of persons with disabilities, a variable required to calculate the OOP payment corresponding to each dependent person. The sample was projected from 2008 to 2012, applying the weights of situations of dependence by level and autonomous community, and considering the 2012 population data.

### 2.2. Levels of Dependence

The first step was to classify the persons with disability into the levels of dependence defined in the DA. The DA defined three levels of dependency: mild (level I), moderate (level II), and severe (level III). To this end, we compared the level of support required to perform basic and instrumental activities of daily living included in the SDDS with the evaluation scale [68] set out in the DA. The final score used to assign the levels of dependence is the result of adding the product of the basic activities of daily living’ tasks requiring support, weighted by the weight of each activity in the overall calculation and the level of support required by each individual. The final score can range from 0 to 100 points: between 0 and 24 points, not eligible; 25–49 points, mild level; 50–74, moderate level, and 75–100, severe level. This methodology is similar to that used in previous studies [69,70] and can be summarized as follows (Equation (1)).
$$\sum_{i=1}^{n}t_{i}s_{i}\left(a_{1i}e-a_{2i}\left(e-1\right)\right),$$
where *i* denotes task for which the individual has difficulty (*n* = 52);*t_i_* = weight of the task;*s_i_* = level of support required by each individual: partial supervision, 0.90; maximum supervision, 0.95, and special supervision, 1;*a*_1_ = weight assigned to the activity in the case of mental illness;*a*_2_ = weight assigned to the activity in the case of no mental illness;e = 0 if the individual does not suffer mental illness;e = 1 if the individual suffers mental illness.

At this point, we established two scenarios. The first scenario was designed as a partial application of DA, including individuals with dependency levels II and III. The second scenario adds individuals with dependency level I, showing a complete application of DA.

### 2.3. OOP Payments Associated with LTC

The second step was to calculate beneficiary’s OOP payments. To do it, previously it is necessary to identify the cost of the service they receive. Services included residential care, day/night care, and home help services. In the case of cash benefits, linked to services, referred to payments to family caregivers and support for non-professional caregivers and personal assistance.

The economic cost of each service is shown in Table 2. We assumed common prices in all the regions of Spain given that the national regulations provide for reference prices for the cost of the services and benefits included in the Act. In this sense, the possible differences across autonomous communities are on the mean. Services were valued in accordance with current legislation [59]. The mean interval of cost was used as defined in the DA for residential care, i.e., €1350/month, subject to an increase of 40% to €1890/month, as set out in the law, in level III cases, which require permanent help for basic activities of daily living. Following the same criterion for day/night care center services, the cost for levels I and II was €650/month, increasing by 25% to €812.5/month for level III cases. Finally, in the valuation of home help benefit, we used the mean interval of hours as defined in the Act [71], which was 10, 33, and 58 h/month for levels I, II, and III, respectively. Regarding the cost per hour of this service, as the distribution of services did not provide information on the number of hours dedicated to personal care and home help, the cost of which was €14/h and €9/h, respectively, we calculated the cost by using the mean cost of both services, i.e., €11.5/h.

Cash benefits are different to services in that the dependent person does not have to pay previously, but rather a transfer of revenue to the household. In order to calculate the cost of economic benefits relating to services and personal care, the end service was considered to usually be residential care, day/night center, and home help. Given the lack of official public statistics, the reference for the service cost was calculated using as weighting factor the weight of the cost of each service in the total cost of services according to level of dependence and autonomous community of residence. To calculate the cost of personal care, home help services were used as the proxy asset.

Regarding cash benefits for family care, the cost was estimated using the cost-replacement method [72,73,74]. Drawing on the information provided by the SDDS-08, we calculated the weekly hours of informal care received by dependent persons according to their level of dependence and autonomous community of residence [75], limited to a maximum of 16 h per day [76,77,78]. In order to value each hour, we used the minimum salary of domestic workers [79], which was €5.02/h. 

The OOP payment of each type of benefit was calculated in accordance with legal definition [59]. Calculations are shown in Table 3. The economic capacity of beneficiaries was calculated in accordance with the criteria set out in the DA, although only an income was taken into account, as the SDDS provides no information on family assets. In the case of benefits for services the corresponding calculation was applied, while for cash benefits, OOP payment was calculated as the difference between the cost of the service (benefit linked to the service and personal care) or the care (benefit for family care and support for non-professional caregivers) and the amount of the assigned benefit, calculated on the basis of the provisions laid out by the Act. 

### 2.4. Impoverishment and Catastrophic Measures

Once the OOP payment is estimated, the aim is to identify to what extent this OOP payment affects the impoverishment or catastrophic expenditure of families. To this end, we used the measures of impoverishment and catastrophe defined by Wagstaff and Van Doorslaer (2003) [34]. The impoverishment rate refers to the number of households whose equivalent income (x_i_) is below the so-called poverty threshold. The equivalent income household was calculated as the relation between the household income and the number of equivalent members or consuming units (n’_i_). To do this, we used the OECD modified equivalence scale [80,81], which assigns a value of 1 to the first household member; 0.5 to each household member aged 14 or over; and 0.3 to each member aged 13 or under. The poverty threshold is defined as a certain level of income, which in this study was fixed as 60% of the mean equivalent household income in Spain, based on the Life Conditions Survey [34,82,83]. The poverty threshold for 2012 was calculated to be €7166/year, i.e., €597.17/month.

To analyze the impoverishing impact of OOP payment, we defined two indicators: the pre-payment poverty rate (H^pre^), that is, before any OOP payment was implemented, and the post-payment poverty rate (H^post^), that is once the OOP payment was implemented. The pre-payment poverty rate (H^pre^) corresponds to households that, before making the OOP payment for dependent care, have an equivalent income below the defined threshold. The post-payment poverty rate (H^post^) was calculated by subtracting the amount of household income, the OOP payment for dependent care, thus establishing their net equivalent income (x’_i_). Our first hypothesis is that OOP payment for dependent care generates a significant impoverishment effect on household finances, i.e., a considerable number of households are situated below the poverty threshold after having made the corresponding OOP payment.

Both indicators, H^pre^ and H^post^, permit a vision of the two aspects of the impoverishing impact of OOP payment of dependent care: incidence and intensity. On one hand, the difference between the poverty rate before and after OOP payment (H^post^)–(H^pre^), shows the incidence of OOP payment, that is, the number of households below the poverty threshold due to OOP payments of dependent care, which had not initially been poor. On the other hand, it shows the increase in the intensity of the poverty gap in households, which were already classed as poor. In this way, a household whose net equivalent income is below the poverty threshold is considered a poor household and the difference between the two incomes is defined as the poverty gap for each household. Thus, the sum of the individual poverty gaps forms the overall poverty gap. Ultimately, we had the overall poverty gap of households that were poor before the OOP payment of dependent care; the poverty gap for households that falling below the poverty threshold due to making dependent care OOP payment (it is to say, the poverty gap due to making the OOP payment); and lastly, the poverty gap of new poor households falling below the poverty threshold due to making the dependent care OOP payment. The last two measures represent the overall poverty gap due to OOP payment of dependent care.

In the same way as health care OOP payments, those for long-term care can also represent a catastrophic expenditure for households if they force individuals or households to suffer a drop in the standards of living now, or in the future [31,84]. The catastrophe threshold (z_cat_) has been defined as a certain percentage of (x_i_), which households must devote to making the corresponding OOP payment for dependent care, (oop_i_), in such a way that when a household has to make a payment above the regulatory percentage, this household is classified as catastrophic. The catastrophe incidence has been defined in terms of the percentage by which the OOP payment of dependent care exceeds the catastrophic threshold (z_cat_), and the mean monthly catastrophic gap has been defined as the amount of income exceeding the catastrophic threshold that is destined to the OOP payment of dependent care. The overall catastrophic gap is the sum of the individual catastrophic gaps.

Although setting a cut-off point is arbitrary since a small payment may be catastrophic for a poor household, in the literature, these thresholds range from 5% to 40% [31,34,35,83,85]. In order to analyze the sensitivity of our calculations the regulatory percentages used were 10%, 20%, 30%, and 40%. Our second hypothesis, in the same sense as the first one, is that OOP payment for dependent care has a significant effect in terms of catastrophism on household finances, i.e., a considerable number of households are situated above each threshold used after having made the corresponding OOP, especially significant in the threshold of 40%, which constrains the household finances.

Both the costs of services and the income of the dependent persons are restated to values in euros of 2012, using the consumer price index as a restatement factor [86]. The monthly income variable was evaluated for each individual and multiple imputations were conducted to estimate missing data [87].

## 3. Results

Table 4 shows the sociodemographic information of the study sample. More than two thirds of the dependent persons were women (68.41%, 67.18%, and 68.08% for levels I, II, and III, respectively). The mean age ranged from 70.74 years (DT: 18.00) for level I to 75.52 (DT: 20.03) for level III. The predominant marital status in persons with level I dependence was married (44.64%), while in levels II and III widowed was the predominant status (42.33% and 48.12%, respectively). In all levels the marital status of separated was the least common. Regarding educational level, for all levels of dependence unfinished primary education was the most common (55.96%, 60.27%, and 66.12%, levels I, II, and III, respectively). Between 79.06% (level I) and 86.97% (level III) of the dependent persons reported their only economic activity to be receiving a pension (contributory or otherwise). The mean number of persons per household was 2.8 in the case of level 1 dependent persons, 2.92 for level II, and 3.11 for level III. These figures were lower in the case of equivalent members (1.82, 1.93, and 2.03, respectively). Finally, mean monthly income ranged from €1375.29 (SD: 1023.72) for grade I to €1516.46 (SD: 1108.97) for level III.

Table 5 shows the mean amount a dependent person has to pay depending on their level of dependence: these amounts ranged from €303.64 in the case of level I to €661.62 for level III. This means that dependent persons pay more than 50% of the cost of their dependent benefit at all three levels. Specifically, level I dependent persons devote a third of their income to this payment (31.85%), level II, almost half (44.83%), and level III, almost two-thirds (64.95%).

The OOP payment of dependent care has an impoverishing impact on dependent persons. Table 6 and Table 7 show both scenarios, partial and complete, and reveal that OOP payment generated an impoverishment rate ranging from 19.16% (level I) to 27.54% (level III); in other words, these are households that newly fall below the poverty threshold after making the OOP payment. Furthermore, we had those households already below the threshold before making the OOP payment (46.27% for level I, and 45.84% for levels II and III). The mean monthly poverty gap per new poor household amounts to €206.52 (SD: 177.97), €283.15 (SD: 171.98), and €507.15 (SD: 249.41) for levels I, II, and III, respectively, while in the case of the initially poor households the poverty gap increases by €286.76 (SD: 144.21) for level I, €451.76 (SD: 170.04) for level II, and €667.10 (SD: 177.64) for level III.

In terms of aggregate values, this implies that the poverty gap due to OOP payment of dependent care of all households below the poverty threshold amounts to €2833.2 million only for levels II and III, and totally for three levels for the second scenario (when level I is included) increase by 26.65%, to €3588.2 million (B + C): of which €963.3 million corresponds to new poor households (C), and €2624.9 to already poor households (B). Consequently, the final poverty gap, including the pre- and post-OOP payment amounts, for the total of the three levels is €5460.5 million (€4115.4 million only for levels II and III). This means that the poverty gap before making the dependent care OOP payment (A) accounted for 31.16% for scenario one and 34.28% for the second scenario of the final poverty gap, 49.65% and 48.08%, respectively for scenarios one and two, of which is due to the increase in the intensity of impoverishment of already poor households after payment. The remaining 19.19% and 17.64% is the gap corresponding to the households that fall below the poverty threshold because of making the OOP payment, for first and second scenarios, respectively. Furthermore, it can be seen that the intensity of impoverishment is higher for households that were already poor before payment (72.12% first scenario and 73.15% second scenario) than for new poor households (27.88% first scenario and 26.85% second scenario). 

Table 8 and Table 9 show the catastrophic expenditure effects of paying dependent care OOP payments. It can be seen that more than 80% of households dedicated more than 10% of their income to dependent care OOP payment, specifically 83.43% in the case of level I, 81.19% for level II, and 97.27% for level III. The mean monthly catastrophe gaps by level are €224.71 (SD: 157.00) for level I, €354.98 (SD: 170.96) for level II, and €526.55 (SD: 239.06) for level III. These percentages and amounts decreased as the cut-off thresholds used increased. Thus, it can be seen that the percentage of households devoting more than 40% of their income to funding dependent care benefits was 23.84% for level I, 51.45% for level II, and 68.07% for level III, with a mean monthly catastrophe gap of €153.78 (SD: 114.48), €172.93 (SD: 119.61), and €341.66 (SD: 137.48), respectively. 

In annual terms, the catastrophe gap generated by devoting more than 10% of the household income to dependent care OOP payment reached €3133.5 million (0.30% of GDP) for scenario 1 (i.e., both levels II and III), and €3955.1 million (0.38% of GDP) for the second scenario (three levels). This amount decreased for levels II and III to €2342.1 million (0.22% of GDP), €1718.9 million (0.18% of GDP), and €1216.3 million (0.11% of GDP) for catastrophe thresholds of 20%, 30%, and 40%, respectively. This values increased when level I was included, to €2821.2 million (0.27% of GDP), €1982.5 million (0.19% of GDP), and €1376.9 million (0.13% of GDP) for catastrophe thresholds of 20%, 30%, and 40%, respectively.

## 4. Discussion

The current work analyzed the economic impact of dependent care OOP payment in Spain on the finances of dependent persons. 

The recession in which Spain has been immersed since 2007 and the fiscal cutbacks implemented have acted as a brake on the expectations for access of dependent persons set out in the 2006 DA [88]. In fact, the modifications to the DA correspond to the third phase of the Spanish financial crisis (2011Q1–2013Q4), which was the period of the most acute economic contraction, characterized by wide-ranging, severe fiscal and structural measures [89]. In this sense, two consequences can be seen: on one hand, there has been a lack of transfer form informal to formal care, and on the other, the OOP payment to be made by dependent persons increased in a scenario of falling income and an excess of household debt, especially in the lowest-earning families [90], increasing impoverishment and fragile health.

Our results show that a dependent person has a mean monthly expenditure ranging from €309.19 for level I dependence to €658.06 for level III. This means that dependent persons are contributing between one- and two-thirds of their income to paying more than 50% of the cost of the services they receive. These figures are far from the initial objectives of the DA, which provided for the cost being proportionally shared by the user, the regional government and the national government. The higher user contribution means that the financial risk for households with dependent persons is increasingly higher.

Our work, in the same vein as studies assessing the impoverishing impact of health care OOP payment, examines the catastrophic and impoverishing effects of OOP payment of long-term care. Expenditure on the OOP payment of long term care has a major impact on households with dependent persons in Spain, since one in five households in the case of level I and one of four in the case of levels II and III (23.13% and 27.54%) fell below the poverty threshold after making the OOP payment in line with our first hypothesis. This impoverishing impact of OOP payment is also found for health care OOP payment in previous studies, albeit in lower percentages [8,31,32,34,35,85].

The impoverishing impact of OOP payment differs according to the level of dependence. The most affected are those with level III dependence, which creates an additional problem since, in the case of level III dependent persons, even though their contributions increase, the demand for the services they receive does not decrease, due to its inelastic nature, and, consequently, certain needs may not be satisfied. In this sense, the years of crisis appear to have increased the barrier to access to healthcare and unsatisfied needs, particularly in the case of persons with less purchasing power [91,92].

In terms of catastrophic expenditure, the pattern of behavior is repeated: more than 80% of households dedicate 10% of their resources to paying the cost of dependence, which represents €3955.1 million (0.38% GDP) if level I is included and €3133.5 only for levels II and III (0.30% GDP). Although as the catastrophe threshold increased, the percentage of households in a catastrophic situation decreased, it was significant that more than half of in the case of level II and two-thirds in that of level III devoted more than 40% of their income to dependent care OOP payment in line with our second hypothesis.

In terms of aggregate values, Spanish households below the poverty threshold pay a total of more than €3588.2 million in dependent care OOP payments. This means that if Spanish public authorities wish to prevent the impoverishment of households with dependent persons, they have to increase their contribution to funding dependent care by that amount.

The likelihood of households with dependent persons falling below the catastrophe threshold may be due to two factors: either because a household with limited economic resources has to make a small payment, or because a household with sufficient resources has to satisfy a large OOP payment. The drafting of the law apparently considered both questions in its provisions for OOP payment. However, the data confirm impoverishment and risk of catastrophic expenditure in households. Drawing on Blomqvist and Busby (2012) [93], the authorities should consider reducing or eliminating OOP payment if users’ incomes fall below a determined level, which could be taken as the poverty threshold. In the same line, certain authors and organizations have suggested that economic policy measures designed to tackle the crisis should take healthcare for vulnerable population groups into consideration [94,95].

Following the legislation implemented in 2012, the funding of dependent care causes a risk of impoverishment and catastrophic expenditure as reflected in the percentages of OOP payments of over 50%. This finding coincides with the results of the study on health care OOP payments conducted by Xu et al. (2007) [8]. Hence, several proposals need to be made. Besides associating OOP payments with the user’s income, exemption form OOP payment should be considered for certain levels of dependence. In addition, thought could be given to the implementation of a funding regime before the situation for dependence arises; in other words, a private health insurance system. As suggested by McIntyre (2006) [32], health insurance could be more appropriate for avoiding the risk of impoverishment. Although funding systems would depend on the institutional structures, culture, traditions, and economic development of each individual country, alternatives need to be found that avoid situations of catastrophic expenditure. In this sense, a limitation to be considered is the way in which the study estimates the cost of care and the corresponding out-of-pocket (OOP) payments related to the cash benefit for family care and support for non-professional caregivers. The use of the value of the time provided by the informal caregiver as the total cost of this cash benefit, despite the maximum value of this cash benefit being recognized in the specific legislation [59,60], implies that the percentage of the user’s income dedicated to OOP payments could be oversized and, consequently, also the results of impoverishment and catastrophism. In a previous study, this same methodology was used in conducting a sensitivity analysis [56] that revealed different results. 

A further limitation of the study is that: while the 2008 Spanish Disability and Dependency Survey was taken as the basis for the socio-demographic characteristics of the dependent population, the copayment for LTC services was calculated according to the 2012 Dependency Act. Nevertheless, the sample was projected from 2008 to 2012, applying the weights of situations of dependence by level and autonomous community, and considering the 2012 population data. Hence, the results should not be greatly affected by this lag in the data sources, since the evolution in the prevalence of disability tends to remain unchanged over time [48]. For instance, in Spain the prevalence of disability amounted to 6.2% in 1999 and 6.5% in 2008, while that of long term care increased only from 4.4% to 5.1% in the same time period [96].

## 5. Conclusions

Our results made various contributions to empirical research on the effect of OOP payment on long-term care in terms of impoverishment and catastrophic expenditure in Spain. The study presents measures of impoverishment and catastrophe related to long-term health care expenditure; it analyzed the consequences of the 2012 OOP payment reform on Spanish households; and it quantified the households that are poorer as a consequence of OOP payment or that are poor following the OOP payment reform and those which the OOP payment has left in a catastrophic situation. 

To the best of our knowledge, this is the first work to analyze the impoverishing impact of dependent care OOP payments in Spain and it contributes to understanding of the economic burden of OOP. The calculation of the monetary amount of OOP payment could be a limitation of the work.

Our results should serve to develop strategies for protection against the financial risk resulting from facing the costs of a situation of dependence. The drafters of future laws might consider a system with exemptions more closely related to families’ incomes. Furthermore, it seems reasonable that political decision makers should implement policies to reduce social imbalance, in this case, alleviating the financial impact on persons with dependence and providing them with services appropriate to their needs. 

## Figures and Tables

**Table 1 ijerph-17-00295-t001:** Increased in out-of-pocket payment associated with long-term care (LTC) in Spain.

		Legislation Pre-Crisis *	Legislation Post-Crisis **	
		Level I/Level II/Level III	Level I/Level II	Level III
In-kind services	Residential Care	70–90% Economic capacity of beneficiary	If monthly income <532.51 ≥ OOP = 0 If monthly income >1541.18 ≥ OOP = 1440	If monthly income < 532.51 ≥ OOP = 0 If monthly income >2117.18 ≥ OOP = 2016
	Day/Night Care Center	10–65% Economic capacity of beneficiary	If monthly income < 532.51 ≥ OOP = 0 If monthly income >1842.28 ≥ OOP = 585	If monthly income < 532.51 ≥ OOP = 0 If monthly income >2207.91 ≥ OOP = 731.25
	Home Help Services	10–65% Economic capacity of beneficiary	If monthly income < 532.51 ≥ OOP = 20 If monthly income >1842.28 ≥ OOP = 567	If monthly income < 532.51 ≥ OOP = 20 If monthly income >2207.91 ≥ OOP = 882
			Level I/Level II	Level III
Cash Benefits	Linked to services	≤60% Economic capacity of beneficiary	** Depends of the cost of the service and the economic capacity of beneficiary	
	For family care and help to support for non professional caregivers	≤75% Economic capacity of beneficiary	If monthly income < 399.38 ≥ OOP = 0 If monthly income >1609.63 ≥ OOP = Maximum, i.e., cash benefit = 0	
	Personal assistance	≤60% Economic capacity of beneficiary	** Depends on the cost of the service and the economic capacity of beneficiary	

* Resolution of 2 December 2008, of the State Secretariat for Social Policy, Families and Dependency and Disability, on the Agreement the Territorial Council for Autonomy and Attention to Dependency, on determining the economic capacity of beneficiaries and the criteria of participation for beneficiaries in the services and benefits provided by the Dependency System [58]. ** Resolution of 13 July 2012, of the State Secretariat for Social Services and Equality, on the Agreement the Territorial Council for Autonomy and Attention to Dependency for the improvement of the system for the autonomy and attention of dependent persons [59].

**Table 2 ijerph-17-00295-t002:** Cost of dependent care benefit by type in euros, 2012.

	LEVEL I	LEVEL II	LEVEL III
Services			
Residential care	Mean interval: €1350/month	Mean interval: €1350/month	Mean interval + 40% increase = €1890/month
Day/night centers	€650/month	€650/month	€650/month + 25% increase = €812.5/month
Home help	Mean hours per month = 10 h Mean cost per hour = €11.5/h 10 × 11.5 = €115/month	Mean hours per month = 33 h Mean cost per hour = €11.5/h 33 × 11.5 = €379.5/month	Mean hours per month = 58 h Mean cost per hour = €11.5/h 58 × 11.5 = €667/month
Cash benefits			
Linked to services	Cost of service = w_1_ × cost of residential care level + w_2_ × cost of day/night centers level+ w_3_ × cost of home help level w_1_, w_2_ y w_3_ = number of benefits service and level and autonomous community of residence/total benefits services level and autonomous community of residence		
For family care and help to support for non professional caregivers	Cost of care = mean number of weekly hours informal care by autonomous community of residence × minimum wage domestic workers (€5.02 h) × 4 weeks/month		
	e.g., national total = 20.95 h/month × €5.02/h × 4 weeks/month = €420.68/month	e.g., national total = 36.89 h/month × €5.02/h × 4 weeks/month = €740.75/month	e.g., national total = 55.63 h/month × €5.02/h × 4 weeks/month = €1117.05/month
Personal assistance	Cost of service = home help service = €115/month	Cost of service = home help service = €379.5/month	Cost of service = home help service = €667/month

**Table 3 ijerph-17-00295-t003:** Beneficiary out-of-pocket (OOP) payments by type of dependent care benefit in euros 2012.

SERVICES	
Residential care	OOP payment = economic capacity beneficiary − MQ MQ (minimum quantity personal expenses= 0.19 × IPREM ^a^)
Day/night centers	OOP payment = (0.4 × economic capacity beneficiary) − (IPREM ^a^/3.33)
Home help	Levels I and II: OOP payment = (0.3333 × cost/h × economic capacity beneficiary) − (0.25 × cost/h) Level III: OOP payment = (0.4 × cost/h × economic capacity beneficiary) − (0.3 × cost/h) Mean cost/h = €11.5/h
Cash benefits	
Linked to services	OOP payment = cost of service − amount of cash benefit assigned Amount of cash benefit assigned = cost of service + MQ − economic capacity beneficiary Level I: 0 < amount of cash benefit assigned < €300/month Level II: 0 < amount of cash benefit assigned < €426.12/month Level III: 0 < amount of cash benefit assigned < €715.07/month
For family care and help to support for non professional caregivers	OOP payment = cost of service − amount of cash benefit assigned Amount of cash benefit assigned = (1.33 × maximum amount of benefit) − (0.44 × economic capacity beneficiary × maximum amount of benefit)/IPREM ^a^ Level I: 0 < amount of cash benefit assigned < €153/month Level II: 0 < amount of cash benefit assigned < €268.79/month Level III: 0 < amount of cash benefit assigned < €387.64/month
Personal assistance	Amount of cash benefit = cost of service + MQ − ECB Cost of service = Cost of home help service CM (minimum quantity personal expenses = 0.19 × IPREM ^a^) Level I: 0 < amount of cash benefit assigned < €300/month Level II: 0 < amount of cash benefit assigned < €426.12/month Level III: 0 < amount of cash benefit assigned < €715.07/month

^a^ IPREM 2012: €532.51/month. IPREM (Indicador Público de Renta de Efectos Múltiples), public multiple effects income indicator, used as a reference for cash benefits and granting financial aid.

**Table 4 ijerph-17-00295-t004:** Sociodemographic variables by level of dependence EDAD-08.

Variables	LEVEL I		LEVEL II		LEVEL III	
**Gender**	*N*	%	*N*	%	*N*	%
Male	115,355	31.59%	134,776	32.82%	88,948	31.92%
Female	249,863	68.41%	275,927	67.18%	189,754	68.08%
**Age (Mean (SD) Min–Max)**	70.74 (18.00) 6–104		72.95 (18.70) 6–104		75.52 (20.03) 6–102	
**Marital status**						
Single	54,846	15.02%	66,100	16,09%	47.495	17.04%
Married	163,029	44.64%	162,967	39,68%	93.142	33.42%
Widowed	135,668	37.15%	173,834	42.33%	134,110	48.12%
Separated	11,675	3.20%	7802	1.90%	3955	1.42%
**Educational level**						
Unfinished Primary	203,721	55.96%	247,144	60.27%	183,435	66.12%
Primary	120,611	33.13%	122,310	29.83%	76,746	27.66%
Secondary	22,939	6.30%	20,091	4.90%	8930	3.22%
Tertiary	16,806	4.62%	20,547	5.01%	8323	3.00%
**Employment status**						
Working	13,587	3.77%	5578	1.38%	1011	0.37%
Unemployed	6865	1.91%	3387	0.84%	1664	0.61%
Receiving pension	284,757	79.06%	350,786	86.60%	236,707	86.97%
Other	54,988	15.27%	45,326	11.19%	32,784	12.05%
**Number of members household (Mean (SD) Min–Max)**	2.8 (1.36) 1–11		2.92 (1.39) 1–9		3.11 (1.34) 1–10	
**Number of equivalent members household (Mean (SD) Min–Max)**	1.82 (0.65) 1–5.2		1.93 (0.66) 1–5		2.03 (0.64) 1–5.3	
**Monthly income (Mean (SD))**	1375.29 (1023.72)		1448.46 (1071.46)		1516.46 (1108.97)	

**Table 5 ijerph-17-00295-t005:** Dependent care OOP payments by level.

	LEVEL I		LEVEL II		LEVEL III	
	Mean	SD	Mean	SD	Mean	SD
Amount	303.64	173.89	412.12	192.71	661.62	221.70
%/total cost benefits	55.23%	18.59%	52.98%	20.56%	52.43%	16.28%
% /household income	31.85%	24.21%	44.83%	35.63%	64.95%	45.61%

**Table 6 ijerph-17-00295-t006:** Impoverishment rate of dependent care OOP payments by level. Scenario 1 (partial application, including levels II and III).

	LEVEL II	LEVEL III	AVERAGE LEVELS II AND III
Poverty rate			
Pre-OOP payment (Hpre)	45.84%	45.84%	45.84%
Post-OOP payment (Hpost)	68.97%	73.38%	70.75%
Increase	23.13%	27.54%	24.91%

Mean household poverty gap (€/month; SD)			
Pre-OOP payment (Hpre pov)	330.90 (277.37)	348.77 (294.87)	338.12 (284.71)
Increase already poor households	451.76 (170.04)	667.10 (177.64)	538.82 (202.85)
Increase poor households due to OOP payments	283.15 (171.98)	507.15 (249.41)	383.25 (237.82)
			TOTAL LEVELS II AND III
“A” (M euros per year)	747,520,920	534,722,160	1,282,243,080
“B” (M euros per year)	1,020,556,800	1,022,768,880	2,043,325,680
“C” (M euros per year)	322,771,200	467,076,120	789,847,320
Increase	1,343,328,000	1,489,845,000	2,833,173,000
A as % (A + B + C)	35.75%	26.41%	31.16%
B as % (A + B + C)	48.81%	50.52%	49.65%
C as % (A + B + C)	15.44%	23.07%	19.19%

B as % (B + C)	75.97%	68.65%	72.12%
C as % (B + C)	24.03%	31.35%	27.88%

**Table 7 ijerph-17-00295-t007:** Impoverishment rate of dependent care OOP payments by level. Scenario 2 (complete application, including levels I, II, and III).

	LEVEL I	AVERAGE THREE LEVELS
Poverty rate		
Pre-OOP payment (Hpre)	46.27%	45.99%
Post-OOP payment (Hpost)	65.44%	68.91%
Increase	19.16%	22.92%

Mean household poverty gap (€/month; SD)		
Pre-OOP payment (Hpre pov)	290.50 (266.86)	321.53 (279.54)

Increase already poor households	286.76 (144.21)	450.99 (220.19)
Increase poor households due to OOP payments	206.52 (177.97)	332.08 (236.18)

		TOTAL THREE LEVELS
“A” (M euros per year)	589,132,680	1,871,375,760
“B” (M euros per year)	581,546,520	2,624,872,200
“C” (M euros per year)	173,440,200	963,287,520
Increase	754,986,720	3,588,159,720

A as % (A + B + C)	43.83%	34.28%
B as % (A + B + C)	43.27%	48.08%
C as % (A + B + C)	12.90%	17.64%

B as % (B + C)	77.03%	73.15%
C as % (B + C)	22.97%	26.85%

**Table 8 ijerph-17-00295-t008:** Catastrophic effect of dependent care OOP payments by level. Scenario 1 (partial application, including levels II and III).

	Catastrophe Threshold (z_cat_, OOP Payment as % of Income)			
	10%	20%	30%	40%
LEVEL II				
% households with catastrophic expenditure (Hcat)	81.19%	72.07%	64.14%	51.45%
Mean monthly catastrophe gap (%; SD)	43.72% (33.80%)	38.80% (32.51%)	32.81% (32.25%)	29.51% (32.67%)
Mean monthly catastrophe gap (euros; SD)	354.98 (170.69)	290.23 (143.52)	221.90 (130.63)	172.93 (119.61)
Overall catastrophe gap (M euros per year)	1,420,472,400	1,030,801,080	701,391,241	438,495,000
LEVEL III				
% households with catastrophic expenditure (Hcat)	97.27%	83.40%	72.14%	68.07%
Mean monthly catastrophe gap (%; SD)	56.54% (45.24%)	55.08% (43.29%)	52.93% (41.33%)	45.84% (40.75%)
Mean monthly catastrophe gap (euros; SD)	526.55 (239.06)	470.13 (205.78)	421.71 (156.62)	341.66 (137.48)
Overall catastrophe gap (M euros per year)	1,713,004,800	1,311,272,400	1,017,454,560	777,759,480
Overall catastrophe gap level II + level III (M euros per year)	3,133,477,200	2,342,073,480	1,718,845,801	1,216,254,480

**Table 9 ijerph-17-00295-t009:** Catastrophic effect of dependent care OOP payments by level. Scenario 2 (complete application, including levels I, II, and III).

	Catastrophe Threshold (z_cat_, OOP Payment as % of Income)			
	10%	20%	30%	40%
LEVEL I				
% households with catastrophic expenditure (Hcat)	83.43%	69.23%	41.68%	23.84%
Mean monthly catastrophe gap (%; SD)	27.06% (23.20%)	21.58% (22.96%)	22.52% (23.84%)	26.61% (22.89%)
Mean monthly catastrophe gap (euros; SD)	224.71 (157.00)	157.91 (145.78)	144.32 (138.88)	153.78 (114.48)
Overall catastrophe gap (M euros per year)	821,657,400	479,111,760	263,643,960	160,647,960
Overall catastrophe gap level I + level II + level III (M euros per year)	3,955,134,600	2,821,185,240	1,982,489,761	1,376,902,440

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
