# Peer review of "Financial Catastrophism Inherent with Out-of-Pocket Payments in Long Term Care for Households: A Latent Impoverishment"

_ijerph, 2020, doi:10.3390/ijerph17010295_

Round 1

Reviewer 1 Report

The estimate of out-of-pocket (OOP) payments (differs significantly from the estimates of the IMSERSO (Government institution) and others made by other authors. Specifically, the authors of the article estimate an annual collection of co-payments of more than double of that estimated by the IMSERSO (€ 3955 M vs. € 1848M) for the year 2012.

One of the main causes of this deviation is how the authors have calculated the cash benefit “for family care and help to support for non-professional caregivers”. The authors do not take the amounts of the benefits approved by the Government for the year 2012, but instead make a calculation according to the number of hours that “should be paid” to the family caregiver. This causes a great deviation between reality and the hypothetical.

The cash benefit “for family care and help to support for non- professional caregivers” in 2012 has a very important weight since it represents almost half of the total benefits (45%) followed by from the services "residential care (13%)" and "home help (13%).

The estimate of the percentage of the OOP payments on the user’s income is oversized by the way of calculating the cost of the benefit “for family care and help to support for non- professional caregivers”.

Other main limitation of the study is said by the authors in the lines 395-399.

Author Response

AUTHORS: We are grateful to the reviewer for this comment. The reviewer is right in indicating the reason why our estimation is higher than that of IMSERSO. Following the reviewer’s suggestion, we have included a new limitation in the corresponding section as follows (lines 416 and 424).

“In this sense, a limitation to be considered is the way in which the study estimates the cost of care and the corresponding out-of-pocket (OOP) payments related to the cash benefit for family care and support for non-professional caregivers. The use of the value of the time provided by the informal caregiver as the total cost of this cash benefit, despite the maximum value of this cash benefit being recognized in the specific legislation [65, 66], implies that the percentage of the user’s income dedicated to OOP payments could be oversized and, consequently, also the results of impoverishment and catastrophism. In a previous study, this same methodology was used in conducting a sensitivity analysis [56] that revealed different results.

A further limitation of the study is that: while […] in the same period [96].”

Reviewer 2 Report

Title: Financial Catastrophism Inherent with Out-of- Pocket Payments in Long Term Care for Households: A Latent Impoverishment.  

At first, I think that a different section with the literature review will enhance the value of the study. The part of literature review should be clear and more extended in order to be understandable the differences of the present study. The authors should give more emphasis on economic meaning of the results at least with one paragraph (see for instance Gkillas et al. 2019) The analysis will be solid incorporating and testing theoretical hypotheses.

I think that a revised version with the abovementioned concerns could be a contribution to the literature.

Literature

k. Gkillas, Vortelinos D., C. Floros, Tsagkanos A., (2019) “Economic news releases and financial markets in South Africa” Economies MDPI. Vol 7(4), 112-125.

Author Response

AUTHORS: We thank the reviewer for this comment. Following the reviewer’s recommendations, we have included the paragraphs below in order to improve the manuscript. Each change is indicated in the text

Firstly, we added the corresponding hypothesis:

(Line 232). “Our first hypothesis is that OOP payment for dependent care generates a significant impoverishment effect on household finances, i.e., a considerable number of households are situated below the poverty threshold after having made the corresponding OOP payment”.

(Line 265). “Our second hypothesis, in the same sense as the first one, is that OOP payment for dependent care has a significant effect in terms of catastrophism on household finances, i.e., a considerable number of households are situated above each threshold used after having made the corresponding OOP, especially significant in the threshold of 40%, which constrains the household finances”.

To respond to the comment on extending the literature.

(Lines 53 and 66). “There is extensive literature examining the impact of out-of-pocket expenditures for health services in overall terms [9-12], and also in specific segments of health services, including drugs [13], interventions [14] or specific diseases, such as cardiovascular diseases [15, 16], HIV [17], cancer [18-21] or non-communicable diseases [22]. In their systematic review, Kiil and Houlberg (2014) [23] found that OOP payment has a negative effect on demand, except in the case of hospitalizations, and reduces the use of prescription medicine. Recently, Yerramilli, Fernández and Thomson (2018) [25] revealed that a number of financial protection studies in Europe focus on middle-income countries, with obsolete information (the last year available in many countries was 2011). In addition, very few studies conduct an in-depth analysis of the effects of catastrophism (incidence and intensity) associated with OOP expenditure, and the factors associated with catastrophism and the likelihood of incurring such catastrophic payments [25].”

(Line 91). “However, to the best of our knowledge, there is no study analysing the catastrophic financial effect of OOP payment associated with LTC in any country.”

Following the reviewer’s recommendations, we have included a new paragraph with greater emphasis on the economic significance of the results (line 109):

“OOP expenditure for LTC involves a significant financial burden for population whose ability to carry out the basic activities of daily living is restricted [61, 62], increasing the risk of financial catastrophism for this vulnerable population group [63, 64].”

Furthermore, in line 374, we have added “in line with our first hypothesis” and in line 389: “in line with our second hypothesis”.

Round 2

Reviewer 1 Report

No comments.